# Roles of Gonadotropin Receptors in Sexual Development of Medaka

**DOI:** 10.3390/cells11030387

**Published:** 2022-01-24

**Authors:** Takeshi Kitano, Tomoaki Takenaka, Hisanori Takagi, Yasutoshi Yoshiura, Yukinori Kazeto, Toshiaki Hirai, Koki Mukai, Ryo Nozu

**Affiliations:** 1Department of Biological Sciences, Graduate School of Science and Technology, Kumamoto University, Kumamoto 860-8555, Japan; himestic@yahoo.co.jp (T.T.); takagi@avss.jp (H.T.); mukai.koki.378@m.kyushu-u.ac.jp (K.M.); ryo.n.g.u@gmail.com (R.N.); 2International Research Center for Agricultural and Environmental Biology, Graduate School of Science and Technology, Kumamoto University, Kumamoto 860-8555, Japan; 3Fisheries Technology Institute, Yashima Station, Japan Fisheries Research and Education Agency, Takamatsu 761-0111, Japan; yoshiura@fra.affrc.go.jp; 4Fisheries Technology Institute, Minamiizu Field Station, Japan Fisheries Research and Education Agency, Shizuoka 415-0156, Japan; kazeto@fra.affrc.go.jp; 5Sanriku Fisheries Research Center, Iwate University, Kamaishi 026-0001, Japan; thiraisf@iwate-u.ac.jp; 6Karatsu Satellite of Aqua-Bioresource Innovation Center, Kyushu University, Saga 847-0132, Japan

**Keywords:** gonadotropin, LH receptor, FSH receptor, TALEN, medaka

## Abstract

The gonadotropins, follicle-stimulating hormone (FSH) and luteinizing hormone (LH), are secreted from the pituitary and bind to the FSH receptor (FSHR) and LH receptor (LHR) to regulate gonadal development in vertebrates. Previously, using *fshr*-knockout (KO) medaka (*Oryzias latipes*), we demonstrated that FSH regulates ovarian development by elevating estrogen levels. However, the *lhr*-KO phenotype in medaka is poorly characterized. Here, we generated *lhr*-KO medaka using the transcription activator-like effector nuclease (TALEN) technique. We analyzed its phenotype and that of *fshr*-KO, *lhr*;*fshr* double-heterozygotes (double-hetero), and double-KO fish. All genetically male medaka displayed normal testes and were fertile, whereas *fshr*-KO and double-KO genetically female fish displayed small ovaries containing many early pre-vitellogenic oocytes and were infertile. Although *lhr*-KO genetically female fish had normal ovaries with full-grown oocytes, ovulation did not occur. Levels of 17α,20β-dihydroxy-4-pregnen-3-one, which is required for meiotic maturation of oocytes and sperm maturation in teleost fish, were significantly decreased in all KO female medaka ovaries except for double-heteros. Further, 17β-estradiol levels in *fshr*-KO and double-KO ovaries were significantly lower than those in double-heteros. These findings indicate that LH is necessary for oocyte maturation and FSH is necessary for follicle development, but that neither are essential for spermatogenesis in medaka.

## 1. Introduction

Gonadotropin-releasing hormone (GnRH) released from the hypothalamus stimulates the pituitary gland to secrete follicle-stimulating hormone (FSH) and luteinizing hormone (LH), two gonadotropins responsible for gonadal development [1,2]. FSH and LH are heterodimers of a common α subunit and different β subunits [3,4]. After secretion from the pituitary gland, these hormones bind to specific receptors, FSH receptor (FSHR) and LH receptor (LHR), in gonadal somatic cells to regulate secretion of sex steroid hormones, such as testosterone, estrogen, and progesterone [5,6,7].

The relationships between sexual maturation and FSHR or LHR functions in vertebrates have been intensively studied. In mammals, FSHR is expressed exclusively in Sertoli cells of the testes and granulosa cells of the ovary [8,9]. *Fshr*-knockout (KO) male mice display small testes and partial failure of spermatogenesis; however, they are fertile [8]. *Fshr*-KO female mice, however, display thin uteri and small ovaries and are infertile because folliculogenesis does not occur before antral follicle formation [8,10]. In fish, Fshr is expressed not only in Sertoli cells and granulosa cells, but also in Leydig cells of the testes and theca cells of the vitellogenic ovarian follicle [11,12,13,14]. Using *fshr*-KO medaka (*Oryzias latipes*), we have recently demonstrated that FSH regulates ovarian development and maintenance, mainly by elevating estrogen levels [15]. *fshr*-KO male medaka show normal testes and are fertile, whereas female medaka display small ovaries and are infertile. These results indicate that the FSH signal via FSHR/Fshr is necessary for ovarian development but is not important for spermatogenesis in mammals and fish.

In mammals, LHR is expressed in Leydig cells of the testes and in theca cells, large follicles, and the corpora lutea of the ovary [16,17]. In *Lhr*-KO male mice, postnatal testes development is blocked, and spermatogenesis is arrested at the round spermatid stage [18,19]. *Lhr*-KO female mice display underdeveloped external genitalia and uteri postnatally [18,19]. Thus, both *Lhr*-KO sexes are infertile [18,19,20]. In fish, Lhr is expressed in Sertoli cells and Leydig cells of the testes and in follicle layers including granulosa cells of the ovary in several fish species, including zebrafish (*Danio rerio*) [14,21,22,23]. In a previous study using *lhβ*-KO or *lhr*-KO male zebrafish, the fertilization rate, sperm motility, and histological structure of the testis were not affected [24]. However, female homozygous *lhβ*-KO fish are infertile, whereas homozygous *lhr*-KO fish are fertile. Therefore, the effects of LHR/Lhr on gonadal development are different between mammals and fish. So et al. [25] reported that FSH specifically activates Fshr, whereas LH can stimulate both Fshr and Lhr in zebrafish, supporting the notion that LH acts through the Fshr in the absence of Lhr, leading to normal fertility in female *lhr*-KO zebrafish. The complex relationship between FSH and LH signaling pathways in fish has complicated elucidation of the roles of these gonadotropic hormones in sexual development. By generating KOs of the gonadotropin receptors in medaka, which are phylogenetically distant from Zebrafish, and analyzing their phenotypes, we can better understand FSH and LH signaling in sexual development and reproduction.

Medaka has desirable features for a model vertebrate, such as easy breeding, short generation time, and small genome, and has been used to study vertebrate reproduction [26]. In addition, transgenic and gene editing methods, using transcription activator-like effector nuclease (TALEN) or clustered regularly interspaced short palindromic repeats (CRISPR)/CRISPR-associated protein 9 (Cas9) technologies, are established [27,28], and the medaka sex-determining gene, *DMY*/*dmrt1bY*, which is located on the Y chromosome, has been identified [29,30,31]. Therefore, medaka is an excellent vertebrate model organism for the molecular genetic analysis of various biological phenomena, including embryonic development and sexual maturation.

In the present study, to reveal gonadotropin receptor function in the sexual development of medaka, we first generated *lhr* single-KO (*lhr*-KO) medaka using the TALEN technique and analyzed its phenotype and sex steroid hormone expression pattern. We then generated *lhr;fshr* double-heterozygotes (double-hetero), *lhr* heterozygote;*fshr* single-KO (*fshr*-KO), and *lhr;fshr* double-KO (double-KO) medaka to investigate the function of the two gonadotropin receptors in gonadal development and reproduction.

## 2. Materials and Methods

### 2.1. Ethics Statement

This study was performed using protocols approved by the Animal Care and Use Committee of Kumamoto University (A2020-014) and according to the ARRIVE guidelines.

### 2.2. Animals

The FLFII medaka line was used in this study. This stock allows the identification of genotypic sex by the appearance of leucophores at 2 days post-fertilization. Fish embryos were maintained in embryo-rearing medium (ERM: 17 mM NaCl, 0.4 mM KCl, 0.27 mM CaCl_2_·2H_2_O, 0.66 mM MgSO_4_, pH 7) at 26 °C under a 14 h light and 10 h dark cycle.

### 2.3. TALEN Preparation

TALEN experiments were performed as described in our recent report [27]. Briefly, the TALEN target sites in the *lhr* locus were searched for using the TALEN Targeter program (https://tale-nt.cac.cornell.edu/node/add/talen, accessed on 10 December 2021), and TAL repeats were assembled using the Golden Gate assembly system and the repeat variable di-residue (RVD) arrays assembly reaction, as described previously [32] but with slight modifications. Resulting plasmids that acquired RVD modules were identified by blue/white selection on IPTG/X-gal LB +ampicillin plates and correct assembly was determined by the size of insert generated by *Bsp*EI restriction enzyme digestion. Synthetic mRNAs were generated using *Apa*I-linearized TALEN expression vectors as templates and the mMessage mMachine T7 Ultra Kit (Ambion) and stored at −80 °C until use.

### 2.4. Microinjection

Microinjection of medaka embryos was performed at the one-cell stage using a Nanoject II (Drummond Scientific Co., Broomall, PA, USA). TALEN mRNAs were simultaneously injected into embryos. After injection, embryos were maintained in ERM at 26 °C in an MIR-153 incubator (SANYO Electric Co., Ltd., Hokkaido, Japan).

### 2.5. Genotyping

The genetic sex of each adult fish was decided by genomic PCR as described previously [33]. PCR was performed using primers specific for *dmy* (DDBJ accession no. AB071534). The conditions were as follows: preheating at 95 °C for 10 min, 40 cycles of PCR at 94 °C for 30 s, 59 °C for 30 s, 72 °C for 1 min, and a final extension at 72 °C for 5 min.

Genotyping of *lhr*-KO, double-hetero, *fshr*-KO, and double-KO medaka was performed by genomic PCR. PCR was performed using specific primers for *lhr* (DDBJ accession no. CP020635; 5′-GCTGAATATCAACAGTCTGAGG-3′ and 5′-CCGTAATTTTCTTAATATCTGTG-3′) and for *fshr* [15]. The conditions were as follows: preheating at 95 °C for 10 min, 40 cycles of PCR at 94 °C for 30 s, 59 °C for 30 s, 72 °C for 1 min, and final extension at 72 °C for 5 min.

### 2.6. Fertility Assessment

The fertility of adult fish was investigated by natural mating with fertile wild-type partners. Fertility assessment was conducted with three mating pairs for each genotype, and individuals that failed to produce fertilized embryos were considered infertile.

### 2.7. Determination of Gonadosomatic Index (GSI)

The adult gonads of double-hetero, *lhr*-KO, *fshr*-KO, and double-KO medaka at approximately 6 months of age were weighed, and the GSI was calculated as follows: GSI (%) = gonad weight (g) × 100/body weight (g).

### 2.8. Measurement of Steroids

Steroid hormones were extracted from gonads of adult individuals in diethyl ether as described previously [34]. The levels of 17α,20β-dihydroxy-4-pregnen-3-one (DHP) in males and females, 11-ketotestosterone (11-KT) in males and 17β-estradiol (E2) in females, were measured using time-resolved fluoroimmunoassays according to the method of Yamada et al. [35]. DHP, 11-KT and E2 were purchased from Sigma Chemicals (St. Louis, MO, USA), and antibodies against E2 and DHP were purchased from Cosmo-Bio Co., Ltd. (Tokyo, Japan).

### 2.9. Histological Analysis of Gonads

The ovaries and testes of adult fish at approximately 6 months of age were fixed in Bouin’s solution at 4 °C overnight. After fixation, each sample was dehydrated, embedded in paraffin, and sectioned serially at a thickness of 5 μm. Then, the sections were stained with hematoxylin and eosin [33]. The number of oocytes per ovary was counted, and the developmental stage of oocytes was determined by size and basic morphological characteristics [36].

### 2.10. Statistics

Experimental results were tested for homogeneity of variance using Levene’s test. Data were analyzed by one-way analysis of variance and then tested with Tukey’s multiple comparison test using SPSS statistics 20 (IBM Corp. Armonk, NY, USA).

## 3. Results

### 3.1. Establishment of lhr-KO Lines

The target sequences of the TALEN for *lhr* and results of gene modification are shown in Figure 1. We obtained medaka with 1, 2, and 10-nucleotide deletions in the *lhr* gene. In the 10-nucleotide deletion, the 133rd leucine residue was replaced with a stop codon by a frameshift. No transmembrane domain or intracellular domain can be synthesized by this transcript because the TALEN was targeted to exon 4, which encodes part of the extracellular ligand-binding domain of Lhr. We mated *lhr* 10-nucleotide deletion heterozygous XY male and XX female medaka to obtain the F1 generation and successfully generated *lhr*-KO (*lhr*^−/−^) medaka in the F2 generation.

### 3.2. Phenotypic Analyses of lhr-KO, Double-Hetero, fshr-KO, and Double-KO Medaka

We first crossed an *lhr* 10-nucleotide deletion heterozygous XX female (*lhr*^+/−^;*fshr*^+/+^) with an *fshr*-KO XY male (*lhr*^+/+^;*fshr*^−/−^). Males and females of double-hetero (*lhr*^+/^^−^;*fshr*^+/^^−^) were selected by genotyping of *lhr* and *fshr* genes, and then they were crossed with each other. This cross generated *lhr*-KO (*lhr*^−/−^;*fshr*^+/+^), double-hetero (*lhr*^+/−^;*fshr*^+/−^), *fshr*-KO (*lhr*^+/−^;*fshr*^−/−^), and double-KO (*lhr*^−/−^;*fshr*^−/−^) medaka, which were genotyped using two-step genomic PCR and HRM analysis. For phenotypic analyses, all genotypically male medaka displayed normal testes containing from spermatocytes to spermatozoa, and no abnormal stage of spermatogenesis was observed (Figure 2A–D), similar to wild-type testes [15]. The double-hetero and *lhr*-KO female medaka also displayed normal ovaries, including full-grown oocytes (Figure 2E,F), similar to wild-type ovaries [15]. In *fshr*-KO and double-KO female fish ovaries, many immature oocytes were observed (Figure 2G,H).

### 3.3. Gonad Development and Fertility in KO Medaka

No significant differences in GSI values were observed in *lhr*-KO and *fshr*-KO males compared with double-hetero male medaka; however, that in double-KO males was significantly decreased (*p* < 0.01) (Figure 3A). In *fshr*-KO and double-KO female fish, GSI values were significantly lower (*p* < 0.01) than in double-hetero female medaka (Figure 3A). However, there was no difference in GSI between double-hetero and *lhr*-KO females.

The number of oocytes per ovary was counted to investigate the developmental stage of ovaries in each genotype (Figure 3B). There were many full-grown oocytes in *lhr*-KO females compared with double-hetero females. Ovary histology was similar between *fshr*-KO and double-KO females, and there were many early pre-vitellogenic oocytes compared with double-hetero fish (Figure 3B).

The fertilization rate of *lhr*-KO males was slightly decreased compared with that of double-hetero male medaka (*p* < 0.05) (Figure 3C). The *fshr*-KO and double-KO male medaka had the same level of fertility as double-hetero males. However, when *lhr*-KO, *fshr*-KO, or a double-KO female was crossed with a wild-type male, fertilized eggs were not obtained (Figure 3D).

### 3.4. Steroid Hormone Levels in KO Medaka

No significant difference in DHP levels was observed among *lhr*-KO, *fshr*-KO, double-KO, and double-hetero male testes at the adult stage (Figure 4A), whereas there was a significant decrease (*p* < 0.01) in *lhr*-KO, *fshr*-KO, and double-KO female ovaries compared with double-hetero female ovaries (Figure 4B). Levels of 11-KT in *lhr*-KO and *fshr*-KO testes were significantly lower (*p* < 0.05, *p* < 0.01) than those in double-hetero testes, while there was no difference between double-KO and double-hetero testes (Figure 4C). E2 levels were the same in *lhr*-KO and double-hetero ovaries, similar to those in wild-type ovaries [15], but were significantly lower in *fshr*-KO and double-KO ovaries (*p* < 0.05, *p* < 0.01) (Figure 4D).

## 4. Discussion

In this study, we generated *lhr*-KO, double-hetero, *fshr*-KO, and *lhr;fshr* double-KO medaka and demonstrated the involvement of Lhr and Fshr in gonad development and fertility of medaka. All genotypically male medaka were fertile and possessed normal testes containing all stages of spermatogenic germ cells. However, in *fshr*-KO and *lhr;fshr* double-KO ovaries, oogenesis halted at the previtellogenic oocyte stage. Furthermore, *lhr*-KO females had full-grown oocytes before maturation, but no ovulation occurred. Consequently, these KO females were infertile. These results strongly indicate that LH and FSH are not involved in male reproduction but play a crucial role in ovarian development and female reproduction in medaka. Furthermore, LH and FSH were shown to have distinct roles in female gonad development; FSH is necessary for the development of follicles, whereas LH is essential for oocyte maturation.

In *lhr*-KO female medaka, ovulation did not occur, although the oocytes developed up to the stage of full-grown follicles. Then, in *lhr*-KO females, levels of DHP, which induces oocyte maturation [37,38], were significantly lower than those in double-hetero female medaka (Figure 4B). It is widely accepted that, in fish, LH induces oocyte maturation by stimulating the production of maturation-inducing hormones (DHP in most fishes) in follicle layer cells [38,39]. Hence, DHP production is suggested to be suppressed in *lhr*-KO female medaka because LH signaling is not established, leading to the inhibition of ovulation. In zebrafish [39] and mammals [40], insulin-like growth factors (IGFs) act as important mediators of LH action on oocyte maturation. Therefore, it appears likely that LH induces the maturation via the IGF system in vertebrates.

The ovarian development phenotype in the loss of Fshr function fish (*fshr*-KO and *lhr;fshr* double-KO) was similar to that observed in our previous study using Fshr-deficient medaka [15]. The present results show that folliculogenesis in *fshr*-KO and *lhr;fshr* double-KO females was halted at the previtellogenic stage (Figure 2G,H). In addition, E2 levels in these females were significantly lower than those in double-hetero females (Figure 4D). Similarly, in zebrafish lacking Fshr, all ovarian follicles arrested at the previtellogenic stage, indicating that FSH/Fshr is crucial for primary growth of oocytes [41]. These findings indicate that the disruption of folliculogenesis can be caused by a decrease in E2 level as a result of deficient FSH signaling. Indeed, in salmonid fish, FSH induces vitellogenesis through elevation of E2 levels [42]. Furthermore, a lack of FSH signaling in female mice causes severe ovarian underdevelopment resulting in chronic estrogen deficiency [10]. Therefore, the mechanism of ovarian development by regulating E2 production via FSH signaling is common in vertebrates. In *Fshr*-KO female mice, early loss of estrogen leads to obesity and skeletal abnormalities that intensify with age [10]. In this study, no obvious obesity and skeletal abnormalities were found in all genotypically female medaka, even though estrogen levels decreased in *fshr*-KO and *lhr;fshr* double-KO female fish (data not shown). Accordingly, roles of estrogen in mammals and teleosts may differ slightly, although further investigation is needed to confirm this.

A similar study has been reported for *lhr* and *fshr* KO zebrafish. Using single *lhr* and *fshr* KO and *lhr*;*fshr* double-KO zebrafish, FSH was shown to be essential for folliculogenesis and LH for oocyte maturation [24,41,43]. *lhr*-KO male and female zebrafish were fertile, whereas double-KO male and female fish were infertile [40], which is not consistent with the present study. So et al. reported that, in zebrafish, FSH specifically activates Fshr, whereas LH can stimulate both Fshr and Lhr [25], meaning that Fshr bind both ligands. This supports the notion that LH signaling acts through the Fshr in the absence of the Lhr, to maintain fertility in *lhr*-KO female zebrafish. In teleosts, the ligand selectivity of Fshr/Lhr is not highly specific [44]; in particular, teleost Fshrs appear to possess a broader but limited functional selectivity for both FSH and LH that might depend on the fish species [45]. However, the present results reveal that females, even *lhr* single-KO females, in which Fshr is active were infertile because ovulation was prevented. This result indicates that ligand selectivity of Fshr/Lhr is apparently exclusive in medaka, as in mammals.

Interestingly, *lhr;fshr* double-KO phenotypes are distinctly different between medaka and zebrafish. In double-KO zebrafish males, testes are underdeveloped, and the fish are infertile [41]; however, in medaka, double-KO males had histologically normal testes with all stages of spermatogenic germ cells, no marked differences in steroid hormone levels compared to double-hetero fish, and fertility was confirmed. In teleosts, both gonadotropins, LH and FSH, directly stimulate gonadal sex steroid hormone production by activating Leydig cells [44,46]. Specifically, 11-KT and DHP, stimulated by FSH, appear to play an important role in spermatogenesis in salmonids [47]. In fact, there are several reports that FSH can induce testicular growth and spermatogenesis in vitro and in vivo [11,48,49]. Based on these observations, the double-KO testis phenotype observed in zebrafish is reasonable. In wild-type male medaka, *fshr* and *lhr* are expressed in all testis stages, from immature to fully mature [50], and it is assumed that gonadotropins are involved. However, even with these functional defects, no phenotype is expressed. To clarify the role of gonadotropins in male medaka, further studies are needed.

## 5. Conclusions

Here, we report the phenotypic analysis of *lhr*- and *lhr;fshr*-KO medaka. The double-hetero, *lhr*-KO, *fshr*-KO, and double-KO male medaka displayed normal testes and fertility. *lhr*-KO female fish, however, showed low levels of DHP, no ovulation, and infertility, although normal ovaries with full-grown oocytes were observed. Moreover, *fshr*-KO and double-KO female fish displayed small ovaries with pre-vitellogenic oocytes and were infertile. We suggest that folliculogenesis was halted at the previtellogenic stage because of deficient FSH signaling that resulted in depleted levels of E2. These findings indicate that, in medaka, LH is necessary for oocyte maturation and FSH is necessary for the development of follicles, whereas these gonadotropins are not essential for male sexual development including spermatogenesis.

## Figures and Tables

**Figure 1 cells-11-00387-f001:**
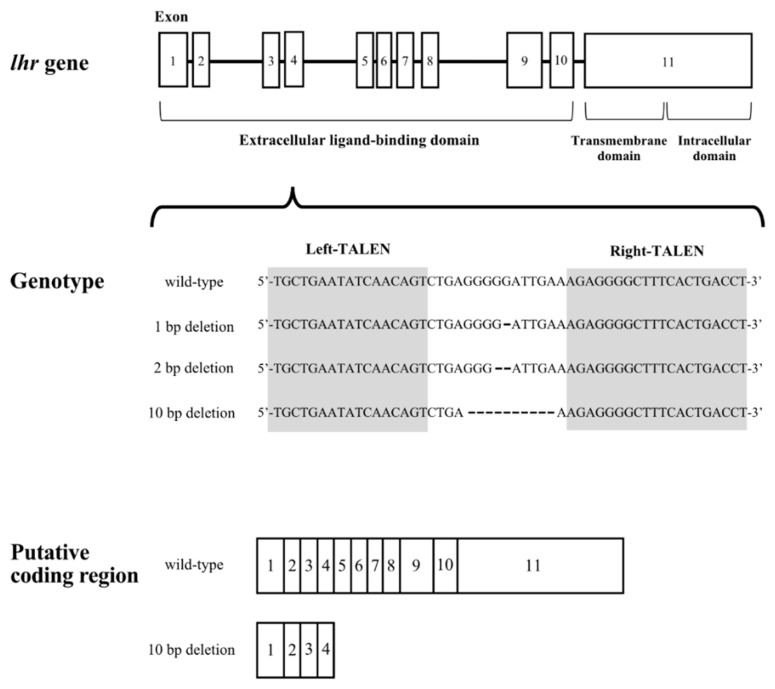
Schematic diagram of the *lhr* gene structure. Gray arabic numerals indicate exon positions. Shaded sequence indicates the sequence targeted by TALEN in exon 4.

**Figure 2 cells-11-00387-f002:**
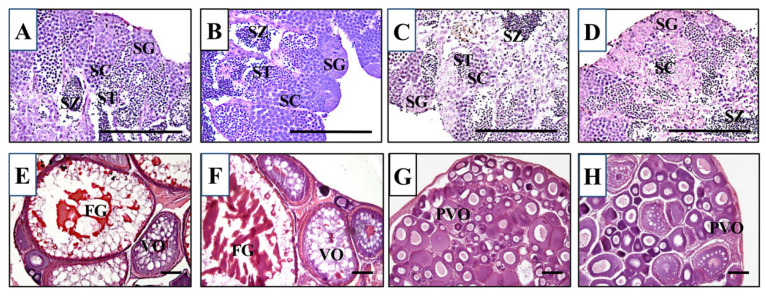
Histological sections of double-hetero (**A**), *lhr*-KO (**B**), *fshr*-KO (**C**), and double-KO (**D**) male medaka testes and double-hetero (**E**), *lhr*-KO (**F**), *fshr*-KO (**G**), and double-KO (**H**) female medaka ovaries. FG: full-grown oocyte, PVO: pre-vitellogenic oocyte, SC: spermatocyte, SG: spermatogonium, ST: spermatid, SZ: spermatozoa, VO: vitellogenic oocyte. Bar, 100 µm.

**Figure 3 cells-11-00387-f003:**
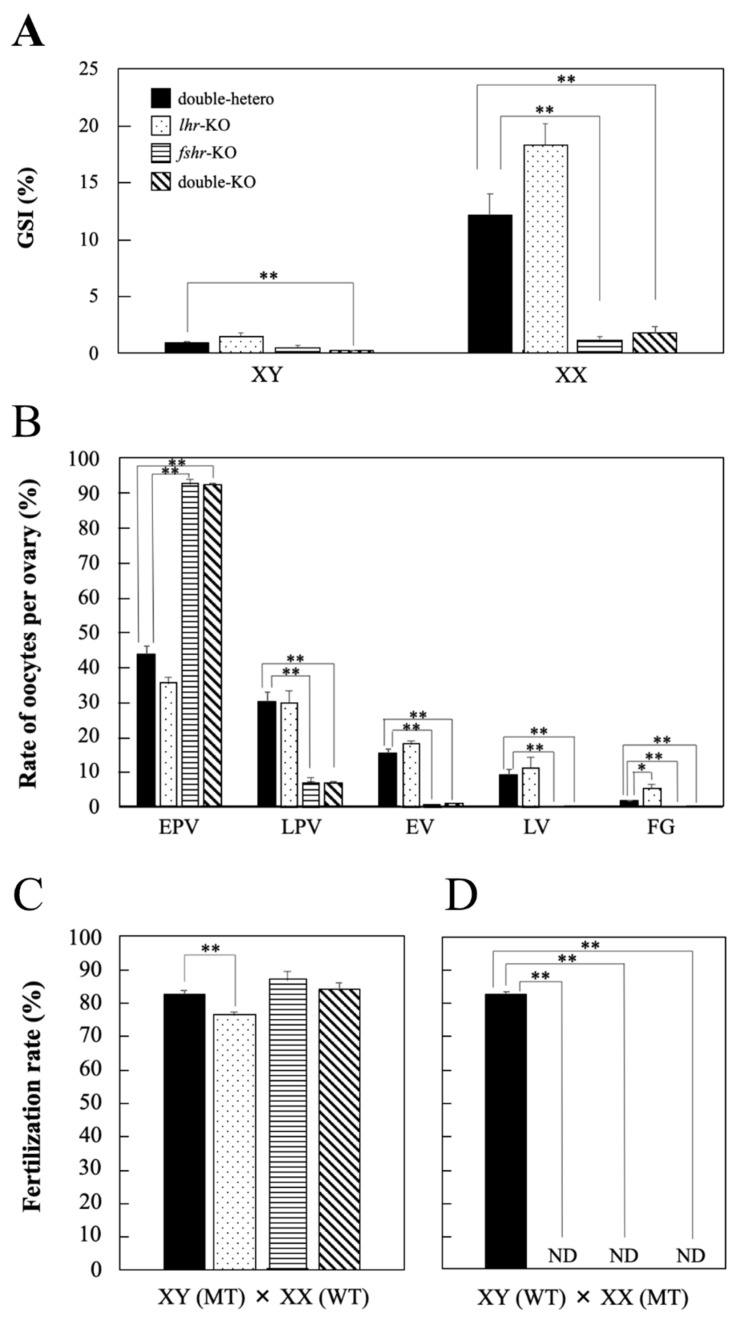
GSI value (%) in gonads of double-hetero, *lhr*-KO, *fshr*-KO, and double-KO adult male (XY) and female (XX) medaka (**A**). The ratio of different stage oocytes in double-hetero, *lhr*-KO, *fshr*-KO, and double-KO female medaka ovaries (**B**). EPV: early pre-vitellogenic oocyte, LPV: late pre-vitellogenic oocyte, EV: early vitellogenic oocyte, LV: late vitellogenic oocyte, FG: full-grown oocyte. Fertilization rate (%) in double-hetero, *lhr*-KO, *fshr*-KO, and double-KO male (**C**) and female (**D**) medaka mated with fertile wild-type partners. MT: mutant type, WT: wild-type. ND: not detected. Individuals that failed to produce fertilized embryos after 3 trials were considered ND. Vertical bar indicates ± SEM (*n* = 3). * *p* < 0.05, ** *p* < 0.01.

**Figure 4 cells-11-00387-f004:**
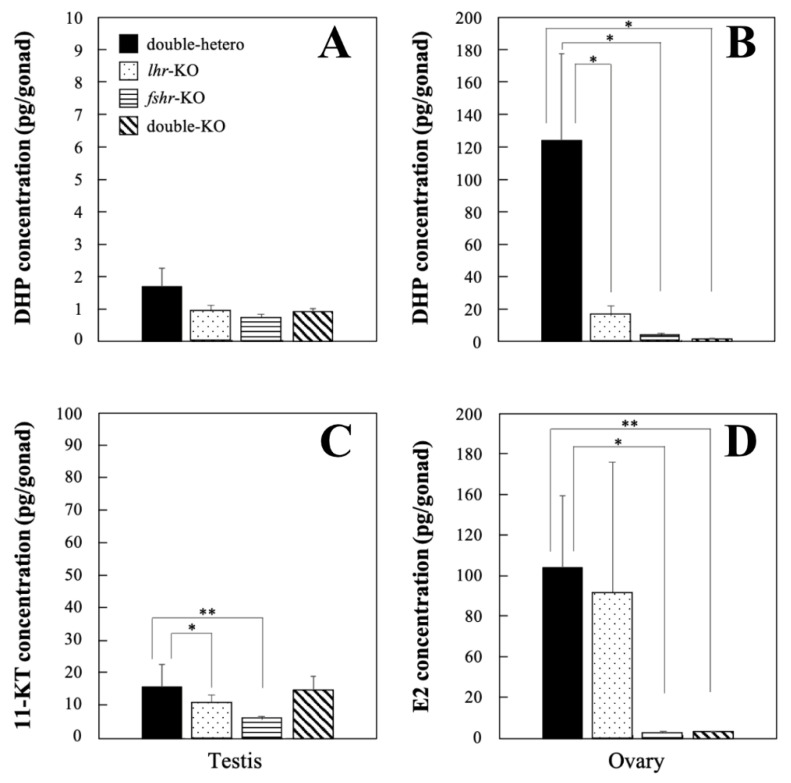
DHP concentration in double-hetero, *lhr*-KO, *fshr*-KO, and double-KO male (**A**) and female (**B)** medaka. 11-KT concentration in male testes (**C**) and E2 concentration in female ovaries (**D**) in double-hetero, *lhr*-KO, *fshr*-KO, and double-KO medaka. Steroid hormones were extracted from gonads of adult medaka. Vertical bar indicates ± SEM (*n* = 3). * *p* < 0.05, ** *p* < 0.01.

## Data Availability

Data are contained within this article. Raw data are available on request from the corresponding author.

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
