# Peer review of "Roles of Gonadotropin Receptors in Sexual Development of Medaka"

_cells, 2022, doi:10.3390/cells11030387_

Round 1

Reviewer 1 Report

This article title “Roles of gonadotropin receptors in sexual development of medaka” was well designed and executed. Even though the roles of FSH and LH receptors is well established in many species, characterization of their roles in medaka was the novel point in this study, which helps in fishery industry. However, there are some minor concerns or future suggestions

This study was limited to phenotypic and reproductive characterization of LH and FSH KO medaka species. There is no or little explanation about the molecular mechanism.

Even though double heterozygous show similar phenotype as that of wildtype, there might be slight difference in the values hormones and other tests. Using wildtype medaka as control was suggested.

Author Response

Dear Reviewer 1,
Thank you very much for your considerate and helpful comments on our manuscript. We revised the manuscript according to your valuable suggestions as described below. We hope that we have responded to all of your comments in a satisfying way. Hope these will make it more acceptable for publication.

Sincerely yours,

Takeshi Kitano

Author’s reply:

Many thanks for the reviewer’s comments. We newly added the explanation about the molecular mechanism in Discussion (lines 266-268). Moreover, we added the phenotypes of wild-type medaka indicated in our paper (Murozumi et al., 2014) in Results (lines 195, 197 and 240).

Reviewer 2 Report

Comments and Suggestions for Authors:

Vertebrate reproduction is controlled by two gonadotropins (FSH and LH) from the pituitary. Despite numerous studies on FSH and LH in fish species, their functions in reproduction remain poorly defined. Interestingly, the authors analyzed that fshr-KO, lhr;fshr double-heterozygotes (double-hetero) and double-KO fish. All genetically male medaka displayed normal testes and were fertile, whereas fshr-KO and double-KO genetically female fish displayed small ovaries containing many early previtellogenic oocytes and were infertile. The manuscript identifies that LH is necessary for oocyte maturation and FSH is necessary for the development of follicles, but that neither is essential for spermatogenesis in medaka

The previous study shows that FSH seemed to play a role in maintaining the female status. Neither the fshb nor lhb mutation alone seemed to affect gonadal differentiation. However, the authors generated lhr-KO medaka using the transcription activator-like effector nuclease (TALEN) technique.

(1) Please the authors need to provide information on which Insulin-like growth factors (IGFs) and inflammation factors are linked to TALEN-mediated gene in medaka of oocyte maturation.

(2) In addition, are there any previous studies on the post-transcriptional regulation of the TALEN technique involved in medaka-related molecular expression? If so, please the authors need to cite the references.

(3) Further understanding of fshr-KO treatment was not just alone affect sexual development but also affect E2 levels was downregulated. But fish oil has been used effectively in the treatment of cardiovascular disease via triglyceride reduction and inflammation modulation. It’s suggested to use more references, such as the papers on the relationship of obesity or hyperlipidemia in humans.

Author Response

Dear Reviewer 2,
Thank you very much for your considerate and helpful comments on our manuscript. We revised the manuscript according to your valuable suggestions as described below. We hope that we have responded to all of your comments in a satisfying way. Hope these will make it more acceptable for publication.

Sincerely yours,

Takeshi Kitano

Author’s reply:

Thank you for your advice. Regarding (1), we cited the papers related to IGFs and mentioned them in Discussion (lines 266-268). Regarding (2), we could not add previous study on the post-transcriptional regulation of the medaka TALEN technique because we could not find them. Moreover, regarding (3), we newly added the information about obesity in mammals and mentioned them in Discussion (lines 281-286).